# Research of Adaptive Extended Kalman Filter-Based SOC Estimator for Frequency Regulation ESS

**Soon-Jong Kwon** [1] , **Gunwoo Kim** [2] , **Jinhyeong Park** [2] , **Ji-Hun Lim** [1] , **Jinhyeok Choi** [1] and **Jonghoon Kim** [2],*

1  Korea Electric Power Corporation Research Institute, Daejeon 34056, Korea;
   soonjong.kwon@kepco.co.kr (S.-J.K.); thedayseye@kepco.co.kr (J.-H.L.); renahren@kepco.co.kr (J.C.)
2  Department of Electrical Engineering, Chungnam National University, Daejeon 34134, Korea;
   qiqu2771@nate.com (G.K.); pig25t@naver.com (J.P.)
*  Correspondence: whdgns0422@cnu.ac.kr; Tel.: +82-42-821-5657

**Abstract:** To achieve frequency regulation, energy-storage systems (ESSs) are systems that monitor and maintain the grid frequency. In South Korea, the total installed capacity of battery ESSs (BESSs) is 376 MW, and these have been employed to achieve frequency regulation since 2015. When the frequency of a power grid is input, accurately estimating the state of charge (SOC) of a battery is difficult because it charges or discharges quickly according to the frequency regulation algorithm. If the SOC of a battery cannot be estimated, the battery can be used in either a high SOC or low SOC. This makes the battery unstable and reduces the safety of the ESS system. Therefore, it is important to precisely estimate the SOC. This paper proposes a technique to estimate the SOC in the test pattern of a frequency regulation ESS using extended Kalman filters. In addition, unlike the conventional extended Kalman filter input with a fixed-error covariance, the SOC is estimated using an adaptive extended Kalman filter (AEKF) whose error covariance is updated according to the input data. Noise is likely to exist in the environment of frequency regulation ESSs, and this makes battery-state estimation more difficult. Therefore, significant noise has been added to the frequency regulation test pattern, and this study compares and verifies the estimation performance of the proposed AEKF and a conventional extended Kalman filter using measurement data with severe noise.

**Keywords:** battery management system; state estimation algorithm; state of charge; frequency regulation; adaptive extended Kalman filter

## 1. Introduction

Battery energy-storage systems (BESSs) are systems that can realize power savings and increased energy efficiency by supplying electric power to users at specified times. They are composed of a battery, battery management system, power conditioning system, and power management system. Further, they are quick-response resources with high-performance, and are capable of supplying a large amount of energy to a power system by outputting instantaneously in a short time [1].

BESSs can be used in a wide variety of applications when linked to the power grid. Since renewable sources such as wind power and solar power generate intermittent power, which is influenced by natural factors, the direct supply of renewable sources to the system can cause problems such as grid instability [2,3]. Therefore, a renewable energy stabilizing ESS is used to limit the power fluctuation of the system when connected to the renewable energy source. In South Korea, high electricity charges are imposed for heavy loads during peak hours, and low rates are applied during low load periods to induce consumers to reduce electricity use. There is a peak shave ESS that charges surplus power

through the ESS at light-duty hours, and which discharges the ESS during heavy-duty hours to reduce power costs [2]. The reference frequency of power systems globally is either 50 Hz or 60 Hz, depending on the country, and the line frequency varies according to the load usage. All electronic devices are manufactured to operate at the reference frequency of 50/60 Hz, so if the line frequency is not maintained at the reference frequency, it may deteriorate the product quality and efficiency [4–7]. The frequency of the power system continues to fluctuate according to the power demand, and the system generation unit adjusts its output to maintain the line frequency. The frequency regulation (FR) ESS is a highly economical operation method because it can partially take the role of a generator that operates for frequency maintaining purposes, thereby reducing the operating cost of the generator [8–12]. When a BESS is used for frequency regulation, it reacts momentarily to sudden frequency drops due to the transient state or micro demand fluctuation, so it can contribute effectively to the stabilization of the line frequency [13–16]. In this study, an algorithm is designed to analyze operation characteristics when the ESS was used for frequency regulation, and a power pattern is generated and analyzed.

The accurate identification of the BESS battery's state of charge (SOC) is critical to ensuring system safety and reliability. When the SOC of a lithium-ion battery is at a high level, lithium ions that lose electrons owing to the oxidation reaction at the cathode electrode are in the state of being reduced to the maximum at the anode electrode. A low SOC indicates that the oxidation reaction occurs at the anode electrode, and the lithium ion moves to the cathode and the reduction reaction occurs [17,18]. At this time, when the battery is used at an SOC that is too high or too low, the battery may encounter stress due to structural instability, and an instantaneous externally generated current may result in over-voltage or over-discharge conditions [19,20]. In addition, because the life of the battery can vary widely depending on the SOC operating area of the battery, it is very important to accurately estimate the SOC and manage the battery in the correct SOC area in order to improve the life of the BESS. There are various methods for estimating the SOC, including the Coulomb counting method, open-circuit voltage (OCV)-based SOC estimation, model-based SOC estimation, and SOC estimation using machine learning. For example, Various methods of SOC estimation were introduced in detail by Zhongbao et al. [21,22]. An equivalent circuit model (ECM) is a good way to analyze the internal resistance characteristics and time constants by electrically modeling the battery. However, the model parameters vary depending on the cathode material of the battery, and the parameters change with temperature and battery aging, which requires a great number of experimentation and analysis. Analysis of ECM models with different chemical materials and temperature was performed by Alexandros et al. [23]. ECM is a vital element for SOC estimation in various techniques such as Luenberger observer, extended Kalman filter, adaptive extended Kalman filter (AEKF), unscented Kalman filter (UKF), particle filter (PF), sliding mode observer (SMO), etc. Model parameters are a very important factor in these techniques, and model parameters may be change with battery aging [21]. At this time, the accuracy of SOC estimation methods can be greatly degraded, so research for estimating capacity and model parameters as well as deterioration is being carried out. The dual extended Kalman filter is typical, and the SOC estimating filter and the capacity estimating filter work together to improve SOC estimation performance, even when battery aging proceeds. Recently, a lot of researches have been conducted to estimate the battery state through various techniques of machine learning. These data driven techniques are methods of analyzing the correlation coefficient from the voltage, current, temperature, deterioration information, etc., acquired from the battery and using the data for computer learning to predict the state of the battery. For example, a study was conducted to estimate the SOC of a battery in Urban Dynamometer Driving Schedule (UDDS) operating patterns using neural network (NN) by LiuWang et al. [24], and a study was conducted to estimate the SOC by designing a deep neural network (DNN) and learning the data by Ephrem et al. [25]. A support vector machine (SVM), which is one of the machine learning regression techniques, was used to estimate the SOC of an EV battery in Reference [26].

In this paper, the extended Kalman filter (EKF) algorithm, which is one of the model-based SOC estimation methods, is used to estimate the SOC. When significant noise is generated in the operation

data of the BESS outputted through the FR ESS algorithm, it is verified that the SOC can be accurately estimated using the EKF algorithm. The SOC estimation performance of the adaptive extended Kalman filter (AEKF), which automatically updates the noise covariance in the algorithm according to the input data, is compared and verified.

## 2. Design of Operation Algorithm for FR ESS and Power Characteristics Analysis

### 2.1. Operation Algorithm Design of Frequency Regulation ESS

When the ESS is operated for frequency regulation in the grid, it is very economical. Korea Electric Power Corporation, Korea's only utility for grid operation, is operating a 376MW as frequency regulation ESS. The frequency changes continuously for a very short time, and in order to maintain this, the ESS is charged and discharged very frequently. It is very difficult to accurately estimate the SOC in such a complex pattern, and it is necessary to find an SOC estimation method that can accurately estimate the SOC of the FR pattern. In this section, the FR algorithm is designed and the power pattern is analyzed for the SOC estimation study. Design of FR algorithm and EKF algorithm was carried out through MATLAB coding, and patterns generated using waveform transform function of MACCOR series 4000 equipment were tested on the battery.

The FR ESS monitors the frequency of the system and is sensitive to changes in frequency. Figure 1 shows the operation algorithm of the FR ESS. When the frequency of the system is normal, the FR ESS operates at steady state to maintain the 60 Hz reference frequency of the system in South Korea [2,9]. A dead frequency band is set and, when the frequency of the system is outside of this range, the ESS is operated to maintain the frequency of the system at 60 Hz. When the grid frequency is in the dead band, the dead band of the SOC required to maintain the SOC is determined, and the suitable operation is performed to maintain the SOC within the operating range. In the event of an accident in which a plant trips, the frequency of the grid system will drop significantly. At this time, an operation of the transient state is performed using the rated power of ESS to recover the frequency. The speed regulation rate (droop) for determining the sensitivity of the ESS power to the frequency change is set differently for the steady state and transient state [2]. When the frequency is recovered after the transient state, the ESS is controlled to return to the steady state. The grid system in South Korea is very stable, and it is rare to experience the transient state in which a plant trips or where the frequency changes rapidly. Therefore, the test pattern for the frequency regulation experiment used in this paper was generated by inputting the frequency when the ESS operates in steady state.

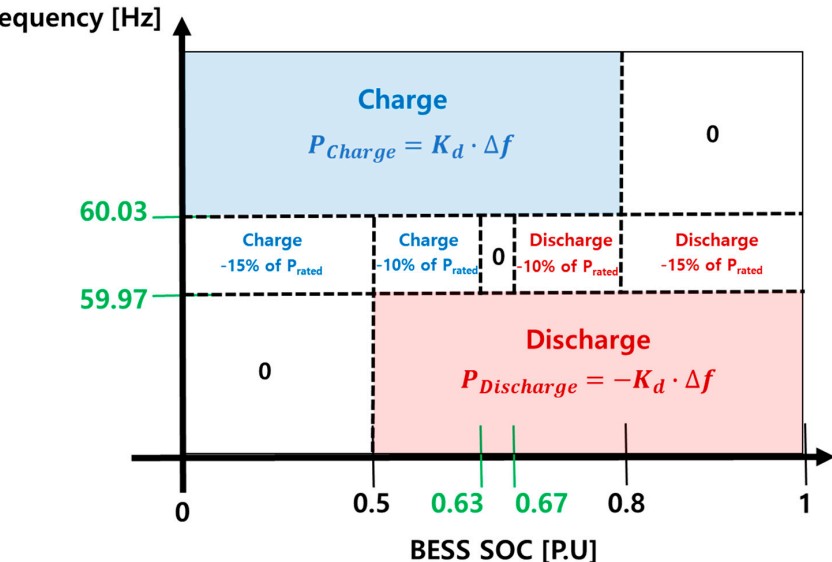

**Figure 1.** Operation algorithm of frequency regulation ESS.

The operator of the ESS decides the *Droop* coefficient to determine how sensitive the ESS will output according to the frequency state. The *Droop* coefficient can be expressed using Equation (1) below.

$$Droop = \frac{\Delta f / f_{ref}}{\Delta P / P_{rated}}. \tag{1}$$

In the above equation, $f_{ref}$ is the reference frequency (60 Hz) and $P_{rated}$ is the maximum power that the ESS can output. The value $\Delta f$ is the difference between the reference frequency $f_{ref}$ and the current system frequency f. The values $f_{ref}$ and $P_{rated}$ are the values already known by the user in the process of constructing the ESS, and when the *droop* coefficient is determined, the proportional coefficient $K_d$ for calculating the power of the ESS can be calculated by applying the *Droop* coefficient as follows:

$$K_d = \frac{\Delta P}{\Delta f} \quad [W/Hz]. \tag{2}$$

If the proportional coefficient $K_d$ is determined, the power of the ESS when the frequency is input can be calculated using Equation (3)

$$P_{ESS} = K_d \cdot \Delta f \quad [W]. \tag{3}$$

$P_{ESS}$ is the power of the ESS when the frequency is input, and is calculated by multiplying the difference between the reference frequency $f_{ref}$ and the incoming system frequency f by $K_d$. If the frequency deviates from the dead band during steady-state operation, the power of the ESS is calculated in the above equation. If the frequency is in the dead band and the SOC is out of the dead band of the SOC, the ESS will charge or discharge a power of 10% (SOC 50–63%, SOC 67–80%) or 15% (SOC 0–50%, SOC 80–100%). The dead frequency band was set to ±0.03 Hz based on 60 Hz and the dead band of the SOC was set to ±2%. The droop coefficient in steady state was set to 0.00279 in order to calculate the ESS power [2].

Figure 2 shows the test pattern of the FR ESS, which was generated by inputting the frequency of the day when the ESS usage was closest to the annual average (normal pattern) and the frequency of the day when the ESS was used heavily (severe pattern), and the results show the experiment performance after applying the test pattern to the battery. The FR pattern of the ESS was generated based on a nickel cobalt manganese battery that has a 20 Ah capacity, and the frequency for one day was set to the test frequency. The algorithm performs power control and applies to the battery the output power obtained as a result of the algorithm. The SOC is calculated by integrating the power, and it can be seen that the pattern operates based on SOC 65%. Table 1 shows the default value of the frequency regulation algorithm used to generate the test patterns in Figure 2.

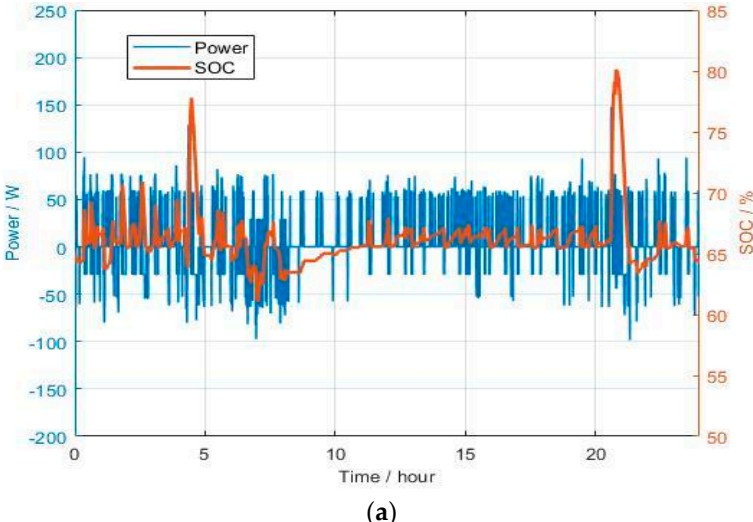

(a)

**Figure 2.** *Cont.*

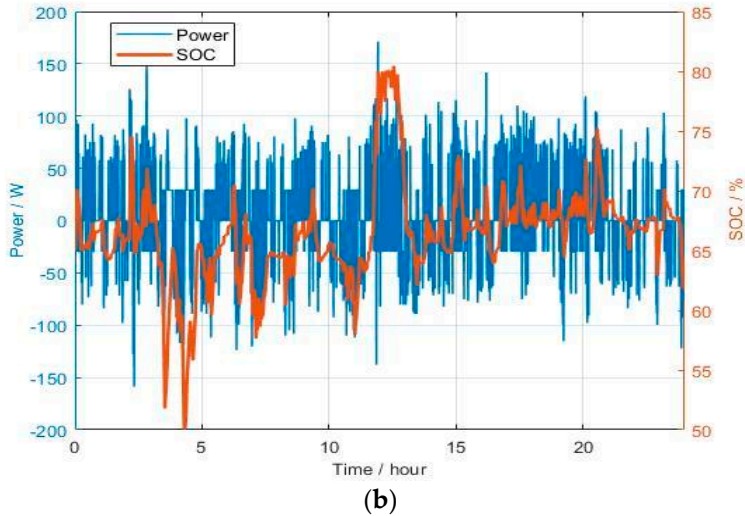

**(b)**

**Figure 2.** Operation result of energy-storage system (ESS) for frequency regulation. (**a**) Normal pattern that used the ESS near annual average; (**b**) Severe pattern that used an ESS that is larger than the annual average.

**Table 1.** Default set value for analysis of frequency regulation algorithm.

| Parameter | Value |
| --- | --- |
| Reference Frequency (Hz) | 60 |
| Dead Band (Hz) | ±0.03 |
| Center SOC (%) | 65 |
| SOC Operation Range (%) | ±2 |
| Rated Power (W) | 292 |
| Nominal Energy (Wh) | 73 |
| SOC Recovery Power "50 < SOC < 80" (%) | 10% (of rated power) |
| SOC Recovery Power "SOC < 50, 80 < SOC" (%) | 15% (of rated power) |

Table 2 compares the result of normal and severe patterns. In the case of the normal pattern, the battery is charged and discharged about 1.57 times per day based on the nominal energy of the battery, and for the severe pattern, it is charged and discharged 3.38 times a day. When the input frequency was 60.03 Hz or higher, the battery charged energy for the frequency regulation was 96.07 Wh in the normal pattern. When the input frequency was 59.97 Hz or less, the energy discharged was 23.40 Wh. In the severe pattern, a frequency of 60.03 Hz or more was reached, and the energy charged by the battery for frequency regulation was 185.76 Wh, which was about 1.9 times higher than in the normal pattern. The energy discharged when the input frequency was below 59.97 Hz was 135.00 Wh, which was about 5.7 times higher than the normal pattern. In the normal pattern, the SOC was less than 63%, and the energy charged for the SOC recovery was 18.62 Wh. When the SOC of battery was 67% or more, the energy discharged was 91.61 Wh. In the severe pattern, the SOC was less than 63%, and the charged energy to recover the SOC was 60.99 Wh, which was about 3.2 times more than the value in the normal pattern. The energy discharged by a SOC of 67% or more was 111.93 Wh, and 1.2 times more energy was discharged compared with the normal pattern. In the case of the normal pattern, the charge operation was performed more often than the discharge operation and, because of the large amount of charge, the operation was performed to discharge the battery to recover the SOC. The maximum power in the test pattern is 147.57 W for the normal pattern and 98.21 W for the discharge. In the severe pattern, 170.94 W is the maximum charge power and 158.73 W is the maximum discharge power. The average C-rate of the battery during charging in the normal pattern is 0.79 and the average C-rate during discharging is 0.45. The average C-rate observed during charging in the severe pattern is 0.68 and average discharge C-rate is 0.59. The normal pattern showed a maximum frequency of

60.08 Hz and a minimum frequency of 59.94 Hz, and the severe pattern showed a maximum frequency of 60.10 Hz and a minimum frequency of 59.91 Hz.

**Table 2.** Analysis results for normal pattern and severe pattern of frequency regulation.

| Parameter | Normal Pattern | Severe Pattern |
|---|---|---|
| Total Energy (Wh) | 229.70 | 493.68 |
| Charge Energy (Wh) | 114.69 | 246.75 |
| Discharge Energy (Wh) | 115.01 | 246.93 |
| Energy for Frequency Regulation "f > 60.03" (Wh) | 96.07 | 185.76 |
| Energy for Frequency Regulation "f < 59.97" (Wh) | 23.40 | 135.00 |
| SOC Recovery Energy "SOC < 63" (Wh) | 18.62 | 60.99 |
| SOC Recovery Energy "SOC > 67" (Wh) | 91.61 | 111.93 |
| Maximum Charge Power (W) | 147.57 | 170.94 |
| Maximum Discharge Power (W) | 98.21 | 158.73 |
| Average Charge C-rate | 0.79 | 0.68 |
| Average Discharge C-rate | 0.45 | 0.59 |
| Maximum SOC (%) | 80.2 | 80.0 |
| Minimum SOC (%) | 61.0 | 50.0 |

The frequency characteristics of the power grid can vary from day to day, depending on the type of load and the time of use. Since the FR ESS operates according to the grid frequency input to the algorithm, the power pattern of the ESS varies for each grid frequency. In addition, since ESS complexly charges and discharges in a very short time, it is very difficult to accurately estimate the SOC of the ESS. The EKF algorithm can accurately estimate the SOC, even for the complex test pattern of the FR ESSs, and can improve the safety and reliability of ESSs.

## 3. Performance Verification of SOC Estimation Using Adaptive Extended Kalman Filter

The operating SOC range of the battery significantly impacts the life of the battery, so it is very important to accurately estimate the current SOC. Various techniques are employed to estimate the SOC, and one of the most popular methods is the Coulomb counting method. The current SOC is calculated by integrating the current measured by the current sensor over time, which is an advantage in that it is easy to implement. However, because the calculation is performed using an open-loop method, the SOC error cannot be corrected, and the noise of the current sensor amplifies the estimation error of the SOC calculated by the Coulomb counting method [27–29]. Another widely used technique is SOC estimation using the SOC-OCV table. By performing experiments, the OCV data with respect to the SOC are obtained and, based on this, the SOC matched to the measured OCV of the battery is estimated relative to the current SOC [30]. This approach is easy to implement, and even if an error occurs in the SOC estimated by the noise of the current sensor, it can be corrected by the SOC according to the OCV, thereby improving the SOC estimation accuracy [28]. However, the accuracy of the SOC-OCV table should be high, and the characteristics of the OCV vary depending on the temperature and degree of deterioration, so it is necessary to secure the SOC-OCV data considering temperature and the state of deterioration. In addition, batteries with flat OCV shapes in the middle SOC region, such as LFP batteries, are not suitable for estimating SOC using only the OCV.

There are also other methods of estimating the SOC using artificial intelligence, such as the use of neural networks, which is a method of estimating the current SOC when arbitrary data are input by learning the measured data of the battery to a computer. Based on the learned data, the accuracy with which the current SOC is calculated by the computer itself may be high, but a long learning time and a large amount of data are required to give a high estimation performance. The method of estimating the SOC using the EKF algorithm can correct the error by using a close-loop method, thereby improving the accuracy of the SOC estimation and estimating the SOC in real time [31,32]. However, because the algorithm operation requires more computation than the open-loop method, there is a relatively

high level of computational hardware, which may increase the system cost. In addition, there is a disadvantage in that the SOC estimation performance can be significantly changed according to the accuracy of the system model in the algorithm. However, the accuracy of the SOC estimation is very high, and it is a method that can accurately estimate the SOC even with external noise [33–35]. In this section, it is shown how to implement the EKF algorithm and estimate the SOC of the battery within the error range. In addition, the SOC estimation result shows that the EKF can correctly estimate the SOC when noise occurs in the input data.

### 3.1. Extended Kalman Filter Algorithm Design

The Kalman filter (KF) is an algorithm that was developed by Rudolf E. Kalman in the early 1960s, and is an algorithm that can perform optimal estimation considering noise when noisy data are input. The KF is a prediction system based on probability theory and it estimates objects with linear motion using the recursive method [27,28,35]. The estimated object should be a normal distribution with an average of $\hat{x}_k$ and an error covariance of $P_k$, as shown in the following Equation (4).

$$x_k \sim N(\hat{x}_k, P_k). \tag{4}$$

The error covariance is an index of the accuracy of the estimated value in the KF. The larger the $P_k$ value, the larger the estimation error, and the smaller the $P_k$ value, the smaller the estimation error. The KF has a system model that shows the correlation between the state variables inside the algorithm, so the unmeasured state variables can be indirectly estimated [34–37]. However, because KFs are designed for linear systems, they cannot be applied to nonlinear systems. Since most systems exhibit nonlinear characteristics, the algorithm developed to solve this problem is the EKF [36]. As the battery exhibits nonlinear characteristics, the EKF is used as the SOC estimation algorithm, and the operation mechanism of the KF algorithm is shown in Figure 3.

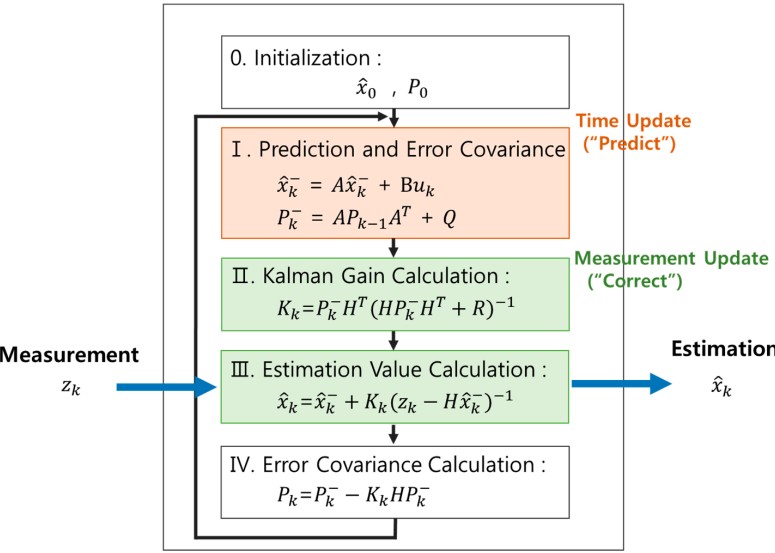

**Figure 3.** Flow diagram of Kalman filter algorithm.

The operating mechanism of the EKF algorithm is similar to that of the KF, but the state variables of the KF are expressed as Equations (5) and (6), considering the nonlinear system model to reflect the nonlinear characteristics.

$$x_{k+1} = f(x_k) + \omega_k. \tag{5}$$

$$y_k = g(x_k) + v_k. \tag{6}$$

The linear matrix of the KF is expressed by a nonlinear function, as shown in Equations (7) and (8).

$$\mathbf{A}x_k \Rightarrow f(x_k). \tag{7}$$

$$\mathbf{H}x_k \Rightarrow h(x_k). \tag{8}$$

The system matrix **A** shows how the system moves over time, and the system matrix **H** shows the relationship between measured values and state variables. The EKF linearizes the nonlinear model to obtain a linear model, which linearizes the model in a way that partially differentiates between the system matrices **A** and **H**. This can be expressed as a Jacobian matrix, as shown in Equations (9) and (10).

$$\mathbf{A} = \left.\frac{\partial f}{\partial x}\right|_{\hat{x}_k}. \tag{9}$$

$$\mathbf{H} = \left.\frac{\partial h}{\partial x}\right|_{\hat{x}_k}. \tag{10}$$

The **A** battery model consisting of one R-C parallel circuit was applied to the EKF algorithm. The values initially input to the EKF algorithm for SOC estimation are the error covariance $P_0$ and initial SOC, as well as the diffusion voltage $V_{diff}$ shown in the RC parallel circuit of the battery model. Activating the algorithm enables us to predict the state of battery and the error covariance. Here, system matrices **A** and **B** for SOC estimation may be expressed as Equations (11) and (12).

$$\mathbf{A} = \begin{bmatrix} 1 & 0 \\ 0 & e^{\frac{-\Delta t}{R_1 C_1}} \end{bmatrix}. \tag{11}$$

$$\mathbf{B} = \begin{bmatrix} \frac{\Delta t}{C_{bat}} \\ R_1\left(1 - e^{\frac{-\Delta t}{R_1 C_1}}\right) \end{bmatrix}. \tag{12}$$

Here, $C_{bat}$ is the capacity of the battery used for the test. When sensor data of the current are input to the algorithm input $u_k$, they are multiplied by the system matrices **A** and **B** to perform SOC and $V_{diff}$ operations of the next time step in order to calculate the predicted value, $\hat{x}_k^-$. The values **Q** and **R** are the noise covariance matrices. **Q** is the noise covariance matrix of the progressive equation, and **R** is the noise covariance for the measured values. Since these two matrices are included in the calculation of the Kalman gain, $K_k$, to calculate the estimated value, the form of the estimated value varies depending on how the values of **Q** and **R** are set. The Kalman gain can be calculated as in Equation (13), and the equation employed to calculate the estimated value, $\hat{x}_k$, in Figure 3 can be expressed differently, as in Equation (14).

$$K_k = \frac{P_k^- H^T}{H P_k^- H^T + R}. \tag{13}$$

$$\hat{x} = (1 - K_k)\hat{x}_k^- + K_k z_k. \tag{14}$$

The value **Q** is used in the calculation of the error covariance, as shown in Equation (15). When **Q** becomes large, the error covariance, $P_{k+1}^-$, becomes large. If $P_{k+1}^-$ becomes large, the Kalman gain calculated in Equation (13) becomes large [38–40], and the Kalman gain is multiplied by the measured data, $z_k$, in Equation (14). As a result, there is a greater influence of the sensor data on the prediction of the battery state. In this case, because the estimate is calculated sensitively according to the measured data, the estimate in which the change is rapid is calculated. In conclusion, when **Q** is small, the estimate is less affected by the measured data, and the estimate for which the change is less is calculated.

$$P_{k+1}^- = A P_k A^T + Q. \tag{15}$$

The value **R** is used for the calculation of the Kalman gain, as in Equation (13), and if **R** is large, the Kalman gain becomes small. Then, the influence of the predicted value, $\hat{x}_k^-$, is greater than the measured value, and the calculated estimated value, $\hat{x}_k$, has a gentle shape. However, if **R** is large, the influence of the predicted value is small, and the influence on the measured value becomes large; the estimated value is then calculated in the form of severe change [36]. This is the opposite to **Q**, and when these features are considered during the algorithm design, the desired result can be obtained. In the EKF, $h(\hat{x}_k^-)$ applying the diffusion voltage calculated as the predicted value, $\hat{x}_k^-$, to the linearized system model can be expressed as in Equation (16).

$$h(\hat{x}_k^-) = OCV_k(SOC_k) + I_k R_0 + V_{diff}. \tag{16}$$

Here, $OCV_k$ is the OCV considering the SOC estimated by the algorithm based on the SOC-OCV data obtained in the experiment. The voltage estimated by the EKF is calculated using the above steps, and the Kalman gain is multiplied by the measured data, $z_k$, as shown in Equation (17). The calculated value is added to the estimated value, $\hat{x}_k^-$, to calculate the estimated value, $\hat{x}_k$.

$$\hat{x}_k = \hat{x}_k^- + K_k(z_k - h(\hat{x}_k^-)). \tag{17}$$

$$P_k = P_k^- + K_k H P_k^-. \tag{18}$$

The Kalman gain is used in Equation (18) to update the error covariance, and the error covariance is used to calculate the error covariance of the next time step in a recursive form. The estimation results of the EKF operating with this mechanism are shown in Figures 4 and 5.

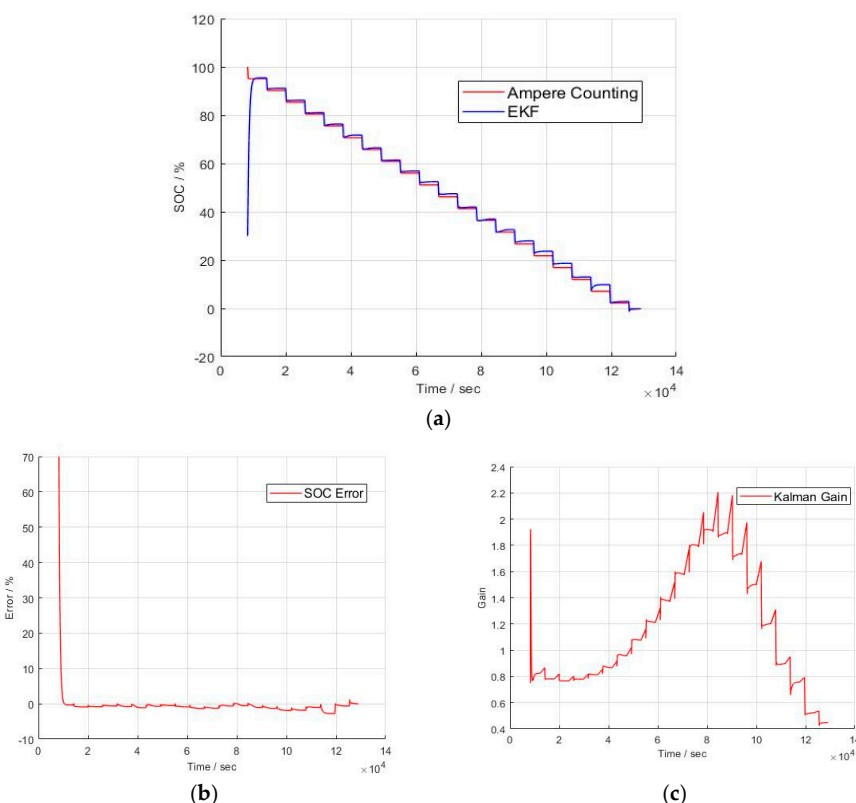

**Figure 4.** SOC estimation of battery using the extended Kalman filter (EKF). (**a**) State of charge (SOC) estimation results of EKF; (**b**) SOC estimation error of EKF; and (**c**) Variation in Kalman gain according to the SOC estimation.

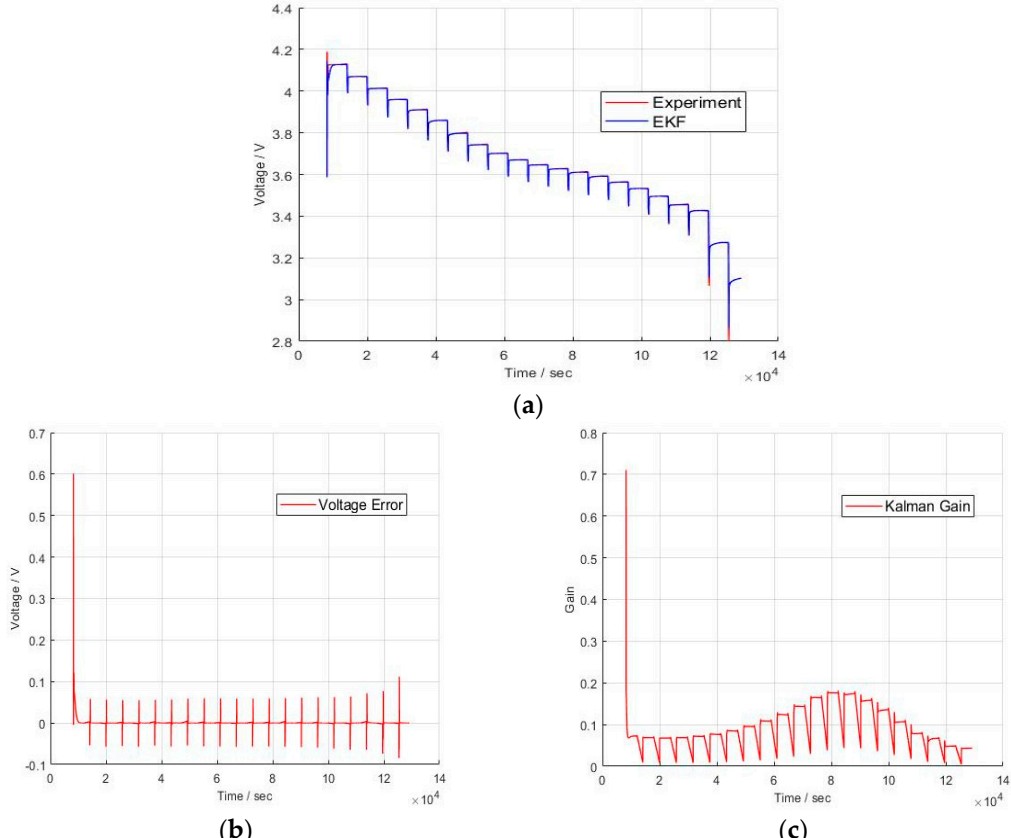

**Figure 5.** Voltage estimation of battery using EKF. (**a**) Voltage estimation results of EKF; (**b**) Voltage estimation error of EKF; and (**c**) Changes in Kalman gain according to voltage estimation.

Figure 4 shows the result of the SOC estimation by the EKF algorithm in the DCIR test in which OCV and equivalent circuit model parameters were extracted by applying a discharge pulse 5% from SOC 100% to SOC 0%. The initial SOC given as the input of the algorithm is set to 30% to confirm whether the SOC is correctly estimated, even if the initial SOC is different from the actual SOC. The estimated SOC can be seen in Figure 4a. Figure 4b shows the SOC estimation error, which shows a large error owing to the error of the initial SOC, but the error is immediately reduced by the EKF algorithm. During the SOC estimation, the algorithm showed an SOC error within 5%. Figure 4c shows the variation of the Kalman gain during the SOC estimation process. As mentioned above, as the Kalman gain increases, the weight of the measured value increases in the calculation of the estimated value, and as the Kalman gain decreases, the weight of the predicted value increases [36]. Based on the variation in the Kalman gain, the Kalman gain is large in the area in which the SOC estimation error is also large. If the estimated value shows an error compared to the measured value, the algorithm automatically increases the Kalman gain of the next time step in order to increase the specific gravity of the measured value and to reduce the estimated error. The error generated during the estimation is corrected for each time step, and the algorithm has an excellent estimation performance.

Figure 5 shows the voltage estimation results obtained for the DCIR test. As the initial SOC input to the algorithm was 30%, the voltage starts with the voltage that matches the SOC, and the error is immediately corrected because it shows an error compared with the measured voltage. As a result of the voltage estimation, the estimation error within 0.1 V is shown as a whole, and the Kalman gain can be confirmed to be close to 0, except for the initial error. Since the SOC estimate is obtained based on the voltage, all estimates of the algorithm are inaccurate unless the algorithm correctly estimates the actual voltage. Therefore, it is very important to accurately estimate the voltage in the EKF algorithm in order to estimate the state of the battery.

### 3.2. SOC Estimation of Frequency Regulation ESS Using Extended Kalman Filter

Assuming that there is no error in the current sensor, SOC estimation using the Coulomb counting method can exhibit a reliable estimation performance. However, all current sensors have minimum error values, and the more expensive and superior the sensor, the smaller the error. Hundreds or thousands of batteries are used in ESSs that operate with high voltage and high current, and the environment is one in which severe noise interference can easily be found in the sensor data. The noise was added as user-defined to the voltage and current data measured by the test equipment to represent the actual noise in the field. At this time, the accuracy of the SOC estimation performance using the Coulomb counting method can be greatly degraded by noise. As the SOC affects the battery stability and cycle life, an accurate SOC estimation is necessary.

In this study, extensive noise was added to the current of the FR test pattern shown in Section 2.1, and the results are presented in this section. In this way, it is verified that the algorithm can estimate the SOC correctly when noisy current data are input to the EKF algorithm. Figure 6 shows the FR test pattern with added noise, and it can be seen that severe noise is generated in the existing current data. When the noise-containing current is input to the algorithm, the SOC estimation result of the algorithm can be seen in Figure 7, and the SOC calculated by the Coulomb counting method with the noise-free current is used as the reference SOC. The SOC estimation result shows that the SOC calculated by the Coulomb counting method with the noise current is different from the reference SOC, whereas the SOC estimated using the EKF algorithm estimates the reference SOC well.

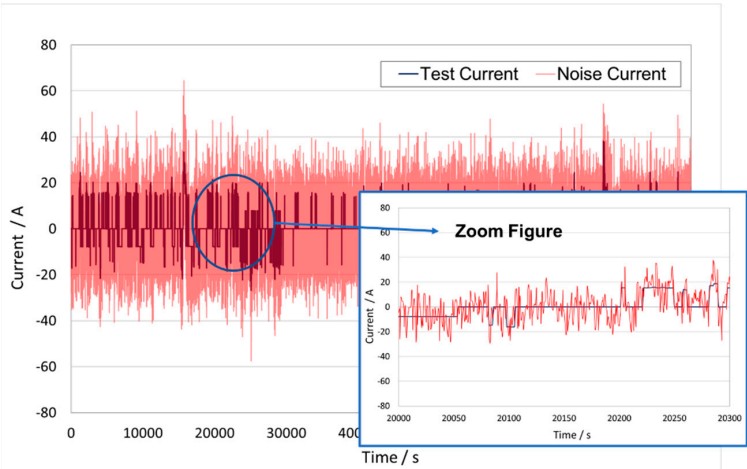

**Figure 6.** Noise current generated in the test pattern of FR ESS.

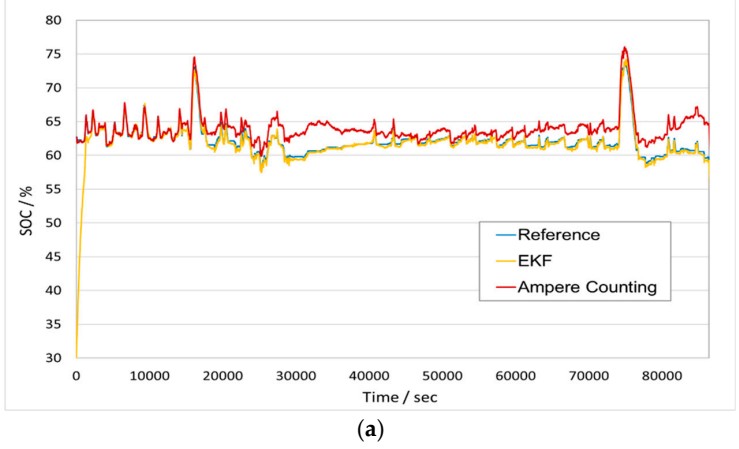

(**a**)

**Figure 7.** *Cont.*

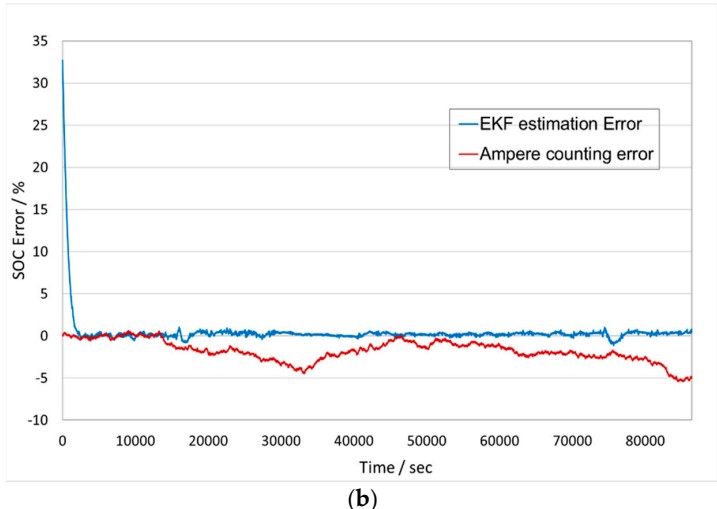

(**b**)

**Figure 7.** SOC estimation in test pattern of FR ESS. (**a**) SOC estimation in test pattern of FR ESS using EKF; (**b**) Comparison of SOC estimation errors of EKF and Coulomb counting methods.

Again, the initial SOC given as the input of the algorithm was 30%, and it was confirmed whether the reference SOC was well estimated even if the initial SOC was different. Although the initial estimation error of the EKF appears to be more than 30%, owing to the initial SOC error, it can be seen that an SOC estimation error of within 3% in the FR test pattern is immediately obtained by the EKF algorithm. The SOC calculation using the Coulomb counting method may show that the error is reduced or is largely generated depending on the shape of the noise. With the Coulomb counting method, for which the error cannot be corrected by the open-loop method, the noise of the current is represented as the error of the SOC, which causes an incorrect SOC calculation. However, the EKF exhibits a high estimation performance because it estimates the SOC considering noise based on the measured voltage even when the noise current is input.

Figure 8 compares the voltage estimation when noise-free current is input to the EKF, and the voltage estimation when noisy current is input to the algorithm. Both results confirm that the measured voltage is well estimated. No noise was added to the voltage measurement data, and only noise was present in the current, so there was no significant effect on the voltage estimation.

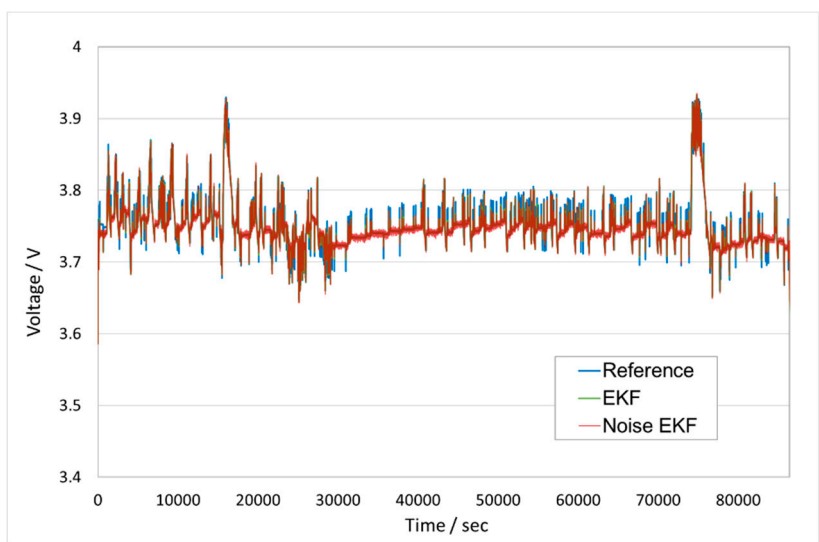

**Figure 8.** Comparison of voltage estimation results in test patterns of FR ESS (EKF vs. EKF that inputs noise current).

### 3.3. SOC Estimation Using Adaptive Extended Kalman Filter

The noise covariance matrices **Q** and **R** of the conventional EKF are input to the algorithm as fixed values on the assumption that the user knows the noise information of the progress equation and the input data. If the noise of the input data is within the expected range, the battery state estimation performance of the EKF algorithm can exhibit high accuracy. However, when the noise is much larger than the set noise, covariance matrices **Q** and **R** occur, and the algorithm cannot properly reflect severe noise information, which may degrade the estimation performance of the algorithm.

The noise covariance matrices **Q** and **R** of the AEKF vary according to the input data, and are recursively input to the algorithm [35,40,41]. The structures of the conventional EKF and the AEKF algorithm are generally very similar. However, in the case of the AEKF, the noise characteristics of the input data can be more actively reflected, and the accuracy of the battery state estimation performance can be obtained in the event of severe noise. M is a window size for moving estimation, and is a monitoring range of data set to reflect noise information of input data in the algorithm. The data with M length is moved and observed in the algorithm, and the noise information **J** is calculated by squaring the estimated error of the output and dividing by M, as shown in Equation (19).

$$\mathbf{J} = \frac{1}{M} \sum_{i=k-M+1}^{k} e_i \cdot e_i^T. \tag{19}$$

**J** has the noise information, and is used to update the noise covariance matrix, **Q**$_{\mathbf{adaptive}}$, of the progression equation and the noise covariance matrix, **R**$_{\mathbf{adaptive}}$, of the measured value, as shown in Equations (20) and (21).

$$\mathbf{Q_{adaptive}} = K_i J K_i^T. \tag{20}$$

$$\mathbf{R_{adaptive}} = J + H_i P_i H_i^T. \tag{21}$$

The AEKF observes the input data and updates the noise covariance matrix, resulting in faster estimation than conventional EKF. Figure 9 shows the results obtained. The SOC estimation performance was not significantly different from that of the conventional EKF, but it can be seen that the AEKF estimates the reference SOC faster than the conventional EKF. These characteristics make it an appropriate algorithm for applications that require fast estimation performance.

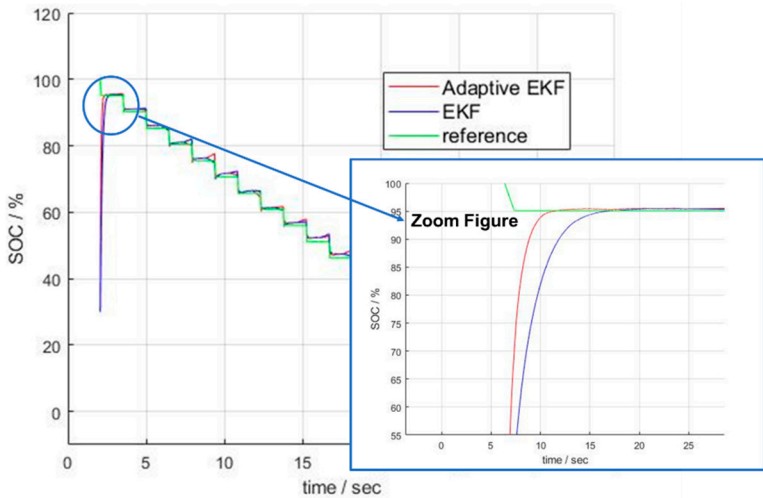

**Figure 9.** SOC estimation comparison of adaptive EKF and EKF.

Figure 10 shows the voltage estimation performance of AEKF when noise is present in the measured voltage data. Figure 10a adds some extensive noise to the voltage data. Figure 10b compares the voltage estimation performance of the EKF and AEKF when the voltage data containing noise are

input, and the conventional EKF has an inaccurate voltage estimation result because it uses fixed noise information, while AEKF shows that the actual voltage is similarly estimated because AEKF updates the fluctuating noise information in the algorithm.

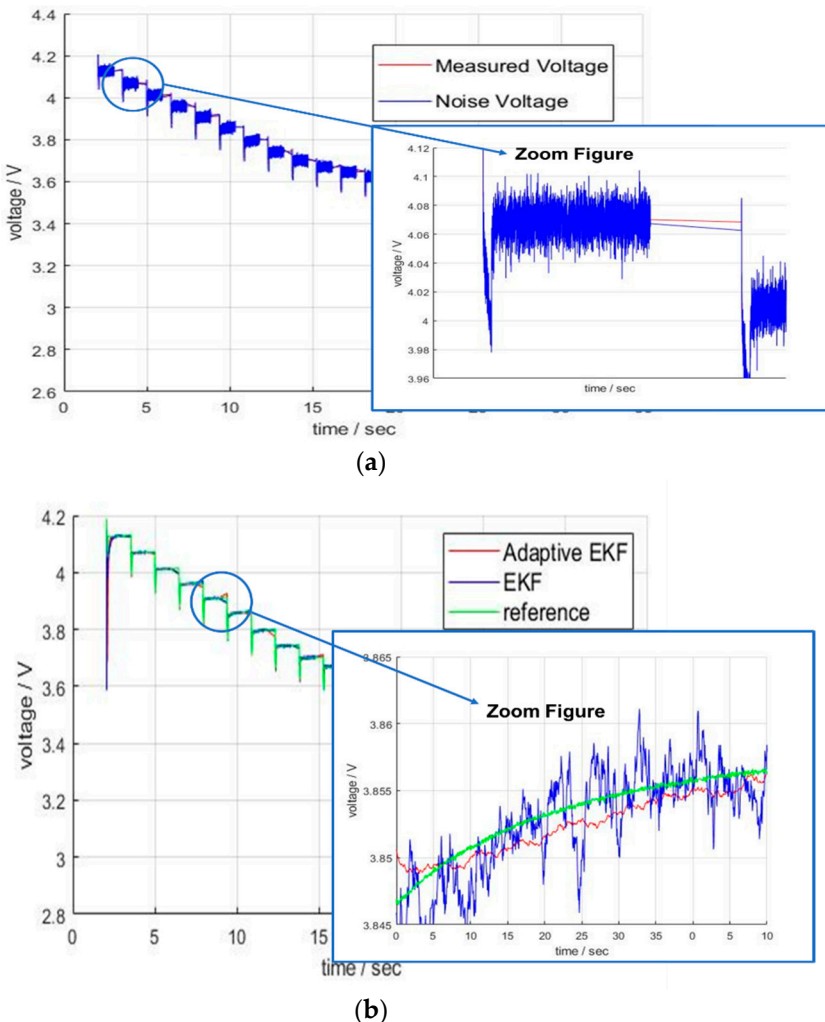

**Figure 10.** Comparison of voltage estimation at noise voltage input. (**a**) Noise in input voltage data; (**b**) Comparison of voltage estimation of AEKF and EKF when noise data is input.

Figure 11 shows the noise covariance matrix, $\mathbf{R}_{\mathbf{adaptive}}$, of the measured data during the battery state estimation of the AEKF algorithm, and the noise covariance matrix, $\mathbf{Q}_{\mathbf{adaptive}}$, of the progression equation. The $\mathbf{R}$ of the conventional EKF algorithm is used for the algorithm calculation in the form of 1 × 1, but in AEKF, $\mathbf{R}_{\mathbf{adaptive}}$ is changed to a 2 × 2 matrix, and is updated as a measurement value is input. It can be seen that the updated $\mathbf{R}_{\mathbf{adaptive}}$ reflects the noise information, and $\mathbf{Q}_{\mathbf{adaptive}}$ is also updated with time. $\mathbf{Q}_2$ and $\mathbf{Q}_3$ are values that remain as 0 from the beginning of the algorithm, and $\mathbf{Q}_1$ and $\mathbf{Q}_4$ vary to reflect noise information of the measurement equation in the algorithm. Since the value of the noise covariance matrix of the AEKF algorithm is updated according to the data input, the estimation performance of the data with the noise component is superior to that of the conventional EKF.

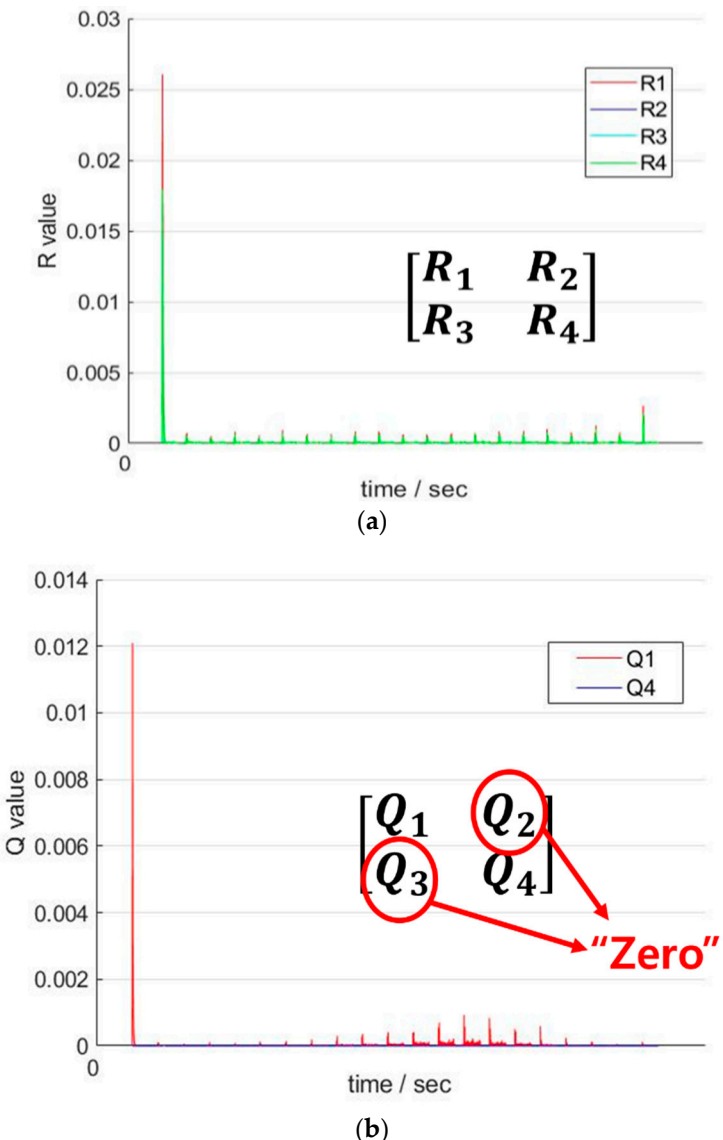

**Figure 11.** Variation of noise covariance matrix during voltage estimation. (**a**) Change in noise covariance $R_{adaptive}$ of measured values; (**b**) Variation of noise covariance $Q_{adaptive}$ in progress equation.

Figure 12 shows the SOC estimation performance of the AEKF algorithm when noise is present in the FR test pattern. Figure 12a shows the noise added to the input current data. Figure 12b compares the SOC estimation performance of EKF and AEKF when these noise currents are input. The initial input SOC of the algorithm is set to 30% in order to confirm that the two algorithms can estimate the reference SOC well. Both algorithms quickly estimate the reference SOC, but it can be seen that the AEKF, which automatically updates the noise covariance matrix, estimates the SOC faster than the EKF. Figure 13 compares the Kalman gain of EKF and AEKF, which change during the SOC estimation process. As the value of the Kalman gain decreases, the algorithm outputs the result by weighting the value estimated by the algorithm rather than the measured value coming into the input. Since noisy data are input, the AEKF is closer to zero than the EKF, and the Kalman gain is lowered. As a result, the AEKF algorithm indicates that there is a lot of noise in the input data, and it can be confirmed that the AEKF algorithm depends more on the estimated value than on the measured value.

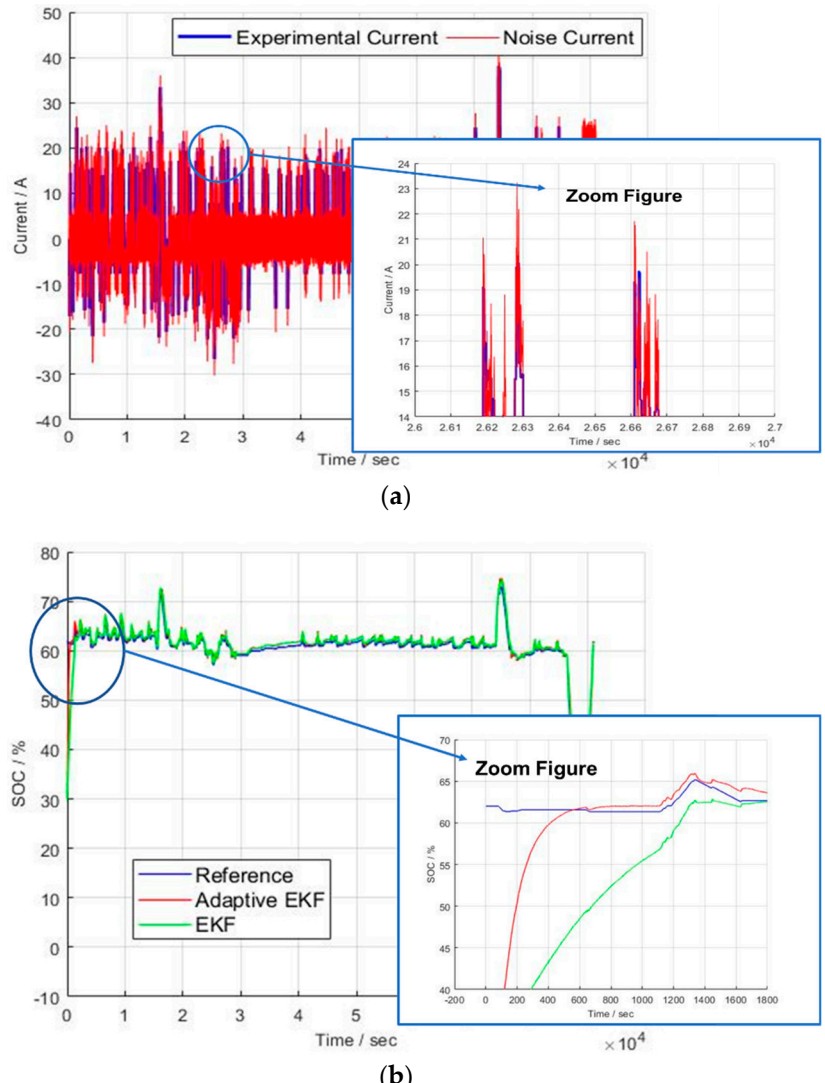

**Figure 12.** SOC estimation of the adaptive extended Kalman filter (AEKF) in a frequency regulation (FR) test pattern; (**a**) Noise generation from input current data; (**b**) SOC estimation comparison of EKF and AEKF.

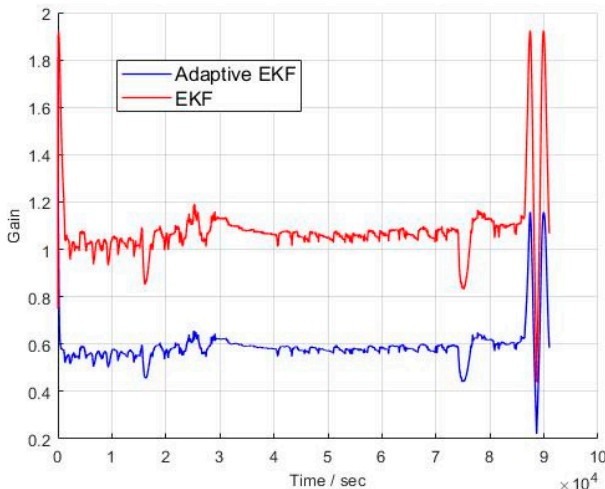

**Figure 13.** Comparison of Kalman gains for EKF and AEKF during SOC estimation.

Figure 14 shows the voltage estimation performance of AEKF when severe noise occurs in voltage data in the FR test pattern. Figure 14a shows the severe noise generated randomly in the voltage data. Figure 14b shows the voltage estimation results of AEKF and EKF. When looking at the estimation result, the EKF estimation result is unstable in the form of noise, although it estimates the actual voltage. This is because the noise covariance of the EKF is used as a fixed value for extensive noise, and the AEKF with the updated noise covariance shows a relatively stable estimation result. Figure 15 shows a comparison of the Kalman gains that change during voltage estimation. The AEKF shows that the Kalman gain converges to near zero owing to the noise of the input data, greatly reducing the weight of the measured value with severe noise. However, because the Kalman gain of EKF has a specific gravity in the measured value relative to AEKF, the noise component of the measured value appears more in the estimated value.

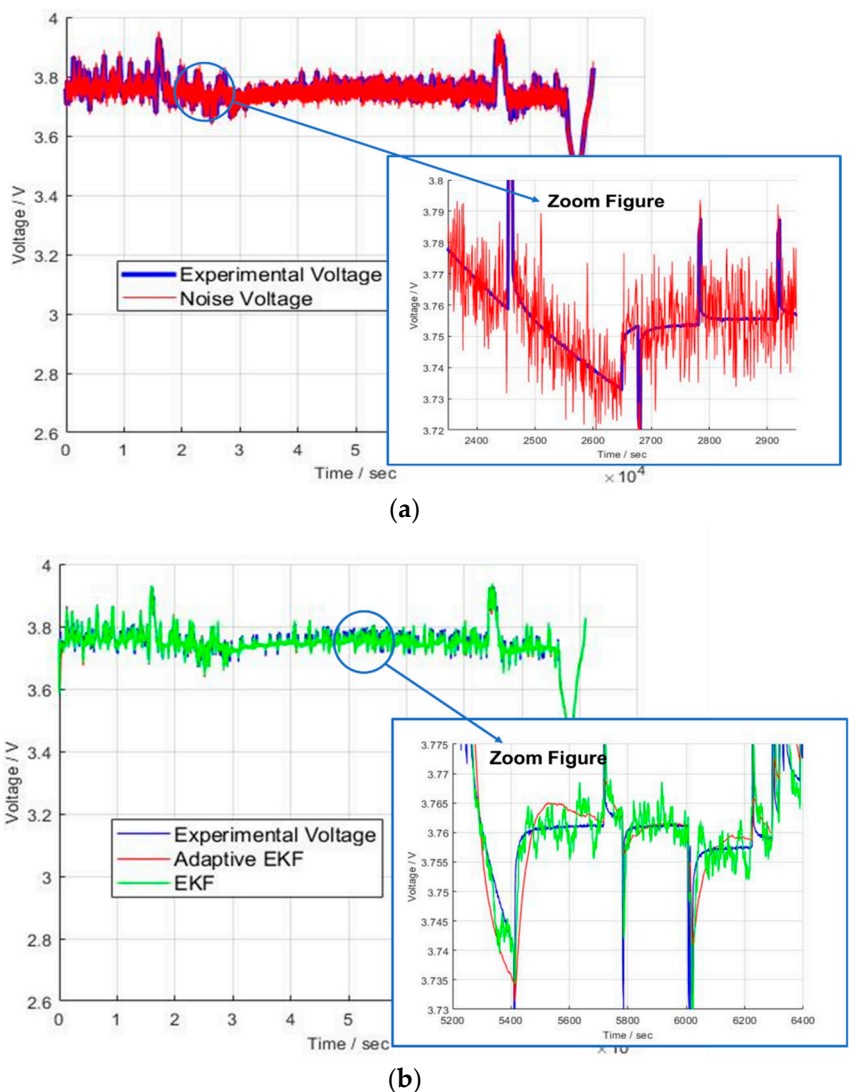

**Figure 14.** Voltage estimation of AEKF in FR test pattern. (**a**) Noise generation from input voltage data; (**b**) Comparison of voltage estimation for EKF and AEKF.

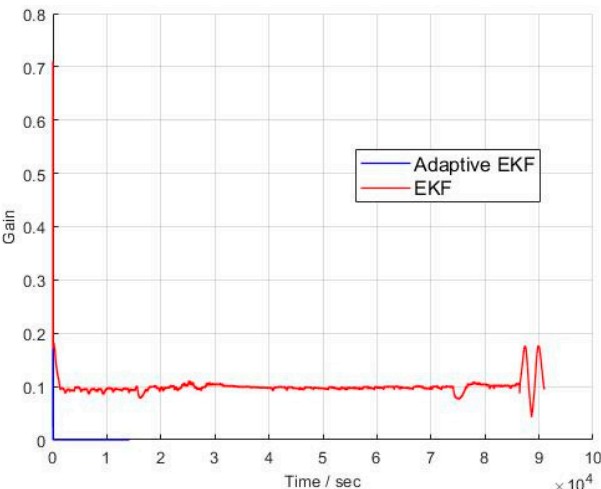

**Figure 15.** Comparison of Kalman gain for EKF and AEKF during voltage estimation.

In this section, it is verified that the EKF can accurately estimate the SOC and voltage, even in the test pattern of the FR ESS, and it is confirmed that the SOC estimation is excellent considering the noise, even when the noisy current data are input. Although the noise covariance matrices **Q** and **R** are used as fixed values in the EKF, the shape of the estimation result varies depending on how the values are set. On the other hand, the AEKF that automatically adjusts the noise covariance matrices **Q** and **R** by considering the noise generated from the input data within the algorithm shows excellent estimation performance for the input data, including noise, compared to the conventional EKF. It has been shown that this is a way of improving the state estimation of the battery.

### 4. Conclusions

In this paper, the FR ESS algorithm was designed and the FR test pattern of the ESS was analyzed to verify the SOC estimation performance of the EKF and AEKF in the test pattern. The frequency on the date on which the ESS usage was close to the annual average, and the frequency on the date when the ESS usage was higher than the annual average, were both obtained from the power grid data. The frequencies for these two dates were input to the FR ESS algorithm to generate a normal pattern and severe patterns, and the operating characteristics of the two patterns were analyzed. The SOC estimation of the frequency adjustment test pattern was performed using the EKF algorithm, and the SOC estimation performance of the EKF was compared with the Coulomb counting method, which is widely used as an SOC estimation technique. Significant noise was also added to the FR test pattern to show the accuracy with which the EKF estimates the SOC for the data that contain noise. In the case of the Coulomb counting method, the SOC estimation performance was greatly degraded owing to the noise of the test pattern, while the EKF algorithm showed an estimation error within 3%. An AEKF which updates the noise covariance according to the incoming input data was designed for comparing and analyzing the SOC and voltage estimation performances compared with the conventional EKF about severe noise data. The AEKF exhibited a better estimation performance than the conventional EKF for the data with severe noise. The AEKF continuously updated the information about noise in the algorithm, and adjusted the severe noise generated in the measured value to the error covariance in the algorithm. By doing this, the estimated value was calculated by placing more weight on the predicted value than the measured data, and it was confirmed that the estimated performance of the AEKF had a better noise elimination than the conventional EKF.

**Author Contributions:** S.-J.K. and J.C. conceived and designed the experiment; S.-J.K., J.-H.L. and J.C. performed the experiment; visualization; J.-H.C., S.-J.K., G.K., and J.P. analyzed the theory; S.-J.K. wrote the manuscript; J.C. and J.K. participated in research plan development and revised the manuscript; supervision J.K.; All authors contributed to the manuscript.

**Funding:** This work was supported by the National Research Foundation of Korea (NRF), grant funded by the Space Core Technology Development Project (No. NRF-2017M1A3A3A03016056) and was supported by the projects of the Korea Electric Power Corporation (R17TA08).

**Conflicts of Interest:** The authors declare no conflicts of interest.

## Abbreviations

The following abbreviations are used in this manuscript:

| | |
|---|---|
| ESS | Energy Storage System |
| BESS | Battery Energy Storage System |
| OCV | Open Circuit Voltage |
| SOC | State Of Charge |
| FR | Frequency Regulation |
| ECM | Equivalent Circuit Model |
| KF | Kalman Filter |
| EKF | Extended Kalman Filter |
| AEKF | Adaptive Extended Kalman Filter |
| UKF | Unscented Kalman Filter |
| PF | Particle Filter |
| SMO | Sliding mode observer |
| UDDS | Urban Dynamometer Driving Schedule |
| NN | Neural Network |
| DNN | Deep Neural Network |
| SVM | Support Vector Machine |

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
