# Peer review of "Research of Adaptive Extended Kalman Filter-Based SOC Estimator for Frequency Regulation ESS"

_applsci, doi:10.3390/app9204274_

Round 1
Reviewer 1 Report
This paper presents the use of AEKF-based SOC estimator for frequency regulation ESS. This paper can be accepted after addressing the following issues:
The literature review for SOC estimation techniques is not enough. Myriads of estimation methods have been proposed over past decade, including the ECM-based method the authors used, eg. Applied energy 204 (2017): 1264-1274; Applied energy 181 (2016): 332-341; IEEE Transactions on Industrial Electronics 66.7: 5724-5735. Such works may be useful to enhance the literature review. More parameters are involved by updating EKF to AEKF, the determination of such parameters should be given for clarification. What is the noise level added to the true measurement? Is it based on real measurement or user-defined?Author Response
Please see the attachment.

Reviewer 2 Report
First of all, I would like to commend the authors on pursuing a very interesting and timely topic. Indeed, researching methods for frequency regulation within energy-storage systems is a worthwhile endeavour. This topic is current and in the scope of the journal Applied Sciences. However, I have some minor considerations.
The Introduction section is quite poor and, partially, the state-of-the art is found in 3 autonomous sections, which is unusual for a research paper. In Section 3, as well as in Section 2, the background represents a large part and it should be found in the beginning of the paper. A section called for eg. "Literature review and hypotheses development" is missing from the paper. In this section, the authors should explain in detail his / her hypotheses, including taking theoretical framework into account.
Because there are many abbreviations / acronyms in the text, a list / table of abbreviations / acronyms would be appropriate.
The authors refer to themselves, within the paper, as Ë®weË® (for eg. Ë®we addË®, Ë®we generateË®… etc.). Please change these with a more formal exprimation (for eg. the authors add, it was generated, etc.).
Line 12 - …(ESSs) are systems that monitor…
Please provide a short description before Figure 1.
The quality of the figures is very poor - they are not legible, there are different fonts used and some of them are too small to be readable. Please provide clearer graphs.
Table 1 is not in accordance with the template. Also, please provide a separate column for the severe pattern's characteristics too. In Table 1, besides the full names, use notations for all characteristics in order to be in accordance with the description in the text and with the formulas before.
Line 145 – 147 - Please provide more information.
In order for the experiment to be reproducible, please provide more information about the software and the framework used to implement and run the algorithm.
To be easier to read, a separation between discussions and partial results would be appropriate in the case of Section 2, respectively Section 3.
Please revise the text, the punctuation, all Figures and Tables, in order to be in accordance with the Journal Template (alignment, fonts, sizes).

Reviewer 3 Report
This paper designed the FR ESS algorithm and analyzed the FR test pattern of the ESS to 397 verify the SOC estimation performance of the EKF and AEKF in the test pattern. This reviewer has concerns on the paper:
1. The contribution of this paper is not very convincing. It seems that the authors only applied known filter to the frequency regulation field.
2. This paper is more like a technical report than a good paper. In addition, the paper seems lack enough theory basis.
3. The format of references is not identical.
Round 2
Reviewer 3 Report
no comment.